# High Glucose Promotes Inflammation and Weakens Placental Defenses against *E. coli* and *S. agalactiae* Infection: Protective Role of Insulin and Metformin

**DOI:** 10.3390/ijms24065243

**Published:** 2023-03-09

**Authors:** Rodrigo Jiménez-Escutia, Donovan Vargas-Alcantar, Pilar Flores-Espinosa, Addy Cecilia Helguera-Repetto, Oscar Villavicencio-Carrisoza, Ismael Mancilla-Herrera, Claudine Irles, Yessica Dorin Torres-Ramos, María Yolotzin Valdespino-Vazquez, Pilar Velázquez-Sánchez, Rodrigo Zamora-Escudero, Marcela Islas-López, Caridad Carranco-Salinas, Lorenza Díaz, Verónica Zaga-Clavellina, Andrea Olmos-Ortiz

**Affiliations:** 1Departamento de Inmunobioquímica, Instituto Nacional de Perinatología Isidro Espinosa de los Reyes, Mexico City 11000, Mexico; 2Posgrado en Ciencias Biológicas, Unidad de Posgrado, Universidad Nacional Autónoma de México, Mexico City 04510, Mexico; 3Posgrado en Ciencias de la Salud, Escuela Superior de Medicina, Instituto Politécnico Nacional, Mexico City 11340, Mexico; 4Departamento de Infectología e Inmunología, Instituto Nacional de Perinatología Isidro Espinosa de los Reyes, Mexico City 11000, Mexico; 5INSERM, UMR 978, Université Sorbonne Paris Nord, UFR SMBH, 93017 Bobigny, France; 6Departamento de Patología, Instituto Nacional de Perinatología Isidro Espinosa de los Reyes, Mexico City 11000, Mexico; 7Departamento de Ginecología y Obstetricia, Hospital Ángeles México, Mexico City 11800, Mexico; 8Ginecología y Obstetricia, Hospital Ángeles Lomas—UNAM, Huixquilucan 52763, Mexico; 9División de Obstetricia, Hospital de Ginecología y Obstetricia No. 4, IMSS, Mexico City 01090, Mexico; 10Departamento de Biología de la Reproducción, Instituto Nacional de Ciencias Médicas y Nutrición Salvador Zubirán, Mexico City 14080, Mexico

**Keywords:** hyperglycemia, hypoglycemics, inflammatory cytokines, bacterial count, bacterial invasiveness, cytokine tolerization, human term placenta

## Abstract

Placentas from gestational diabetes mellitus (GDM) patients undergo significant metabolic and immunologic adaptations due to hyperglycemia, which results in an exacerbated synthesis of proinflammatory cytokines and an increased risk for infections. Insulin or metformin are clinically indicated for the treatment of GDM; however, there is limited information about the immunomodulatory activity of these drugs in the human placenta, especially in the context of maternal infections. Our objective was to study the role of insulin and metformin in the placental inflammatory response and innate defense against common etiopathological agents of pregnancy bacterial infections, such as *E. coli* and *S. agalactiae*, in a hyperglycemic environment. Term placental explants were cultivated with glucose (10 and 50 mM), insulin (50–500 nM) or metformin (125–500 µM) for 48 h, and then they were challenged with live bacteria (1 × 10^5^ CFU/mL). We evaluated the inflammatory cytokine secretion, beta defensins production, bacterial count and bacterial tissue invasiveness after 4–8 h of infection. Our results showed that a GDM-associated hyperglycemic environment induced an inflammatory response and a decreased beta defensins synthesis unable to restrain bacterial infection. Notably, both insulin and metformin exerted anti-inflammatory effects under hyperglycemic infectious and non-infectious scenarios. Moreover, both drugs fortified placental barrier defenses, resulting in reduced *E. coli* counts, as well as decreased *S. agalactiae* and *E. coli* invasiveness of placental villous trees. Remarkably, the double challenge of high glucose and infection provoked a pathogen-specific attenuated placental inflammatory response in the hyperglycemic condition, mainly denoted by reduced TNF-α and IL-6 secretion after *S. agalactiae* infection and by IL-1β after *E. coli* infection. Altogether, these results suggest that metabolically uncontrolled GDM mothers develop diverse immune placental alterations, which may help to explain their increased vulnerability to bacterial pathogens.

## 1. Introduction

Gestational diabetes mellitus (GDM) is a transitory condition of pregnancy characterized by hyperglycemia and low-grade sterile chronic metabolic inflammation. This metainflammatory condition prevails in sera, adipose tissue and, importantly, the placenta [1,2]. Placentas from complicated GDM mothers undergo significant metabolic and immunologic adaptations due to hyperglycemia and metainflammation [2], such as the altered infiltration of immune cells into villous trees [3,4] and increased gene expression for stress- and inflammatory –related genes [5]. Additionally, in vitro studies in a first trimester trophoblast cell line have demonstrated that hyperglycemia induces the secretion of diverse inflammatory cytokines, including interleukin (IL) -1β, IL-6 and IL-8 [6,7]. Despite this high-glucose-dependent proinflammatory response, GDM has been associated with an increased risk of infections, such as vulvovaginal candidiasis, chorioamnionitis and vaginal infections, which are known to contribute to adverse pregnancy outcomes [8]. Likewise, more oral anaerobic bacteria, tuberculosis bacilli, black-pigmented bacteria and actinomycetes have been detected in pregnant women with GDM as compared with nondiabetic pregnant women [9]. 

GDM patients have higher vulvovaginal infection rates and vaginal dysbiosis rates in comparison to euglycemic women [8,10]. Epidemiologic evidence supports that, in diabetic and GDM pregnant women, the most frequent bacterium isolated from urogenital specimens is *S. agalactiae* [11,12], with a colonization-adjusted rate in pregnant women of 21–25% in North America and 18% worldwide [13]. Additionally, *Escherichia coli* is the leading cause of chorioamnionitis and urinary tract infections in pregnant women [14,15,16,17,18]. 

After the diagnosis of hyperglycemia, a pregnant woman is referred to diet counseling, and if the blood glucose levels still exceed the target, metformin and/or insulin are usually indicated [19,20,21,22]. Several studies have widely studied both hypoglycemic drugs in terms of glucose metabolic control, insulin resistance and the prevention of fetal and maternal adverse outcomes [23,24,25]. Additionally, the anti-inflammatory effect of insulin and metformin has been known for several years, as reported among in vivo, in vitro and clinical experiments; being the main underlying mechanism the reduction in inflammatory mediators such as tumor necrosis factor alpha (TNF-α), IL-1β, IL-6 and IL-8, as well as nuclear factor kappa B (NFκB) activation [26,27,28,29,30,31,32,33,34,35,36]. 

Regarding the role of insulin and metformin in the defense against infections, a murine pre-diabetic model revealed that urinary tract infections are more frequent when the insulin receptor is deleted [37]. Additionally, insulin treatment promoted the synthesis of the antimicrobial peptide human beta defensin (HBD) −1 in kidney and colon cell lines [38]. On the other hand, metformin has been described as an antimicrobial agent which reduces bacterial, parasitic and viral infections [39], and recently, its efficacy was demonstrated against uropathogenic *E. coli* [40]. Nevertheless, there is scarce information about the immunomodulatory activities of insulin and metformin in GDM, especially in the context of maternal infections. 

In the present work, we showed that severe hyperglycemia promoted the placental synthesis of inflammatory cytokines, reduced the synthesis of beta defensins and compromised the placental innate defense against *E. coli* and *S. agalactiae* infection. Insulin and metformin treatments proved to not only be effective anti-inflammatories, but they also helped to confront *E. coli* and *S. agalactiae* infection by diminishing bacterial growth and their invasiveness. Finally, the double challenge of high glucose and infection led to a state of immunological unresponsiveness, which may help to explain the increased maternal vulnerability to bacterial pathogens during hyperglycemic pregnancies.

## 2. Results

A total of 35 placentas were processed and analyzed. Clinical data from mothers and newborns are presented in Table 1. The mothers were normotensive, with a pre-gestational BMI < 30 kg/m^2^ and a maternal weight gain < 20 kg, and they had term newborns with a normal weight, length and head circumference. The newborn sex proportion was evenly balanced. 

### 2.1. A Hyperglycemic Model in Human Term Placenta

We evaluated the production of inflammatory cytokines by placental explants in response to a glucose curve (10, 25, 35 and 50 mM). The glucose dose–response curve was chosen according to previous experiments developed in trophoblasts and other cells to mimic normo- and hyperglycemic environments associated with GDM or diabetes [6,43,44,45,46]. 

Treatment with 50 mM glucose during 48 h induced a pro-inflammatory phenotype characterized by a significant increased secretion of TNF-α, IL-1β and IL-6 in comparison to glucose 10 mM (Figure 1A–C). Regarding IL-1β, this cytokine was significantly stimulated starting from 35 mM glucose treatment. Notably, the achieved glucose-dependent increase in inflammatory cytokines was not as high as that observed with lipopolysaccharide (LPS), which represents, in this model, an acute stimulus for inflammation (Figure 1A–C). Considering all the above, we chose 10 mM glucose as a non-inflammatory glucose control, whereas 50 mM glucose was chosen as a hyperglycemic metainflammatory-like condition. Moreover, additional controls for osmolarity were performed to check if the pro-inflammatory stimulus in our placental explant model is exclusively associated with the direct effects of a high glucose concentration, instead of a high osmolar pressure (Appendix A).

The placenta, as a critical organ which needs to efficiently mobilize glucose between the fetus and the mother, functions as a large reservoir of glycogen [47]. Therefore, we evaluated the placental glycogen deposition by Periodic acid–Schiff stain (PAS) as magenta signals. Microphotographs showed that the explants treated with control glucose presented predominant glycogen deposits in perivascular areas and below the basal membrane of syncytiotrophoblasts (Figure 1D, depicted with arrows). In comparison, the explants treated with 50 mM glucose had a broad glycogen distribution throughout the villous mesenchyme, indicating more abundant glycogen deposits (Figure 1E). 

Additionally, we performed an XTT assay with placental explants incubated with control and high glucose concentrations for 96 h. None of the glucose concentrations tested compromised placental viability (Figure 1F), neither insulin nor metformin treatment (Appendix A). Therefore, we confirmed that exposition to 50 mM glucose is a good strategy for emulating a model of hyperglycemia in human term placental explants.

### 2.2. Insulin and Metformin Diminish Placental Pro-Inflammatory Cytokines Secretion under a Hyperglycemic Condition

Considering that insulin and metformin are frequently used as pharmacotherapy in pregnant women with GDM, we wanted to evaluate the effect of both hypoglycemics on the synthesis of pro-inflammatory cytokines in human placenta. Figure 2A–C show that 50 mM glucose significantly induced TNF-α, IL-1β and IL-6 placental secretion. Interestingly, 500 nM insulin added to the hyperglycemic culture media significantly diminished TNF-α secretion (Figure 2A), while this insulin treatment and all the concentrations of metformin tested significantly reduced IL-1β placental secretion under the hyperglycemic culture media (Figure 2B). Metformin was not able to reduce TNF-α secretion. Finally, both treatments were effective in significantly reducing placental IL-6 secretion (Figure 2C). Therefore, insulin and metformin are useful for reducing the hyperglycemic-dependent inflammatory cytokines in the placenta. 

In addition to cytokines, several adipokines are also dysregulated in the serum of GDM women as part of the metainflammatory state [48,49]. Therefore, we also evaluated if glucose and hypoglycemics could modulate placental adipokine secretion (Appendix A). In this model, severe hyperglycemia did not modify the secretion of the main inflammatory (chemerin and leptin) and anti-inflammatory (adiponectin) placental adipokines. Nevertheless, visfatin, an important adipokine with microbicide activity [50,51], was significantly downregulated by hyperglycemia. Intriguingly, insulin treatment further diminished visfatin secretion in comparison with 50 mM glucose, while metformin had no effect upon this adipokine. Hence, in this model, hyperglycemia and hypoglycemics do not modify placental adipokine secretion, except for visfatin, which was downregulated. 

### 2.3. Hyperglycemia Diminishes Placental Beta Defensins Synthesis. Metformin and Insulin Do Not Restore HBDs

We wanted to explore in this model if hyperglycemia and hypoglycemic treatments could modulate the placental synthesis of human beta defensins, as potent microbicides involved in the control of an infection. As shown in Figure 3A–D, severe hyperglycemia significantly diminished the placental production of HBD 1, 2, 3 and 4 in comparison with the explants treated with 10 mM glucose. These observations suggest a weaker innate defense capacity of the hyperglycemic placenta due to the downregulation of at least these four defensins as well as visfatin, although other antimicrobial peptides not evaluated herein could also be affected.

Notably, the co-treatment with insulin or metformin did not reverse the downregulating effect of high glucose upon the production of beta defensins. Therefore, these hypoglycemic drugs did not improve placental innate defense through the direct regulation of HBDs synthesis.

### 2.4. Innate Defense against Bacterial Infections in Pre-Exposed Hyperglycemic Placenta

After we corroborated that severe hyperglycemia results in the higher placental expression of inflammatory cytokines and the lower synthesis of antimicrobial HBDs and visfatin, we hypothesized that these high-glucose-dependent changes could derive in a weakened host immune defense against an infection. To test this, we challenged cotyledon explants with a live Gram-positive (*S. agalactiae*) and Gram-negative (*E. coli*) infection. Additionally, we explored if hypoglycemics could modulate the immune response against these pathogens, considering that they exerted an anti-inflammatory activity. 

First, placental explants were preincubated for 48 h with glucose and hypoglycemic treatments. Then, they were challenged with *S. agalactiae* infection for 4 and 8 h. Afterward, we evaluated bacterial internalization in placental villi by Gram staining (visualized as purple/blue bodies/structures), bacterial growth, and cytokines release in culture media. As shown in Figure 4A, in explants pretreated with 10 mM glucose, bacteria were virtually absent after 8 h of *S. agalactiae* infection. In contrast, pretreatment with 50 mM glucose resulted in a higher proportion of bacteria mostly contained at the syncytial barrier. On the other hand, under a hyperglycemic state, insulin pretreatment qualitatively diminished bacterial invasiveness into placental villi compared with explants pretreated with 50 mM glucose alone, whereas metformin pretreatment did not improve the defense capacity of placenta against *S. agalactiae* infection. Next, we quantitatively analyzed bacterial growth in the culture media. As seen in Figure 4B, pretreatment with 50 mM glucose led to a higher number of *S. agalactiae* at 4 and 8 h post infection in comparison with the explants pretreated with 10 mM glucose. However, neither insulin nor metformin pretreatment modified bacterial counts in comparison with the explants exposed to 50 mM glucose. Altogether, these observations may indicate that insulin does not exert a microbicide effect in the placenta (as observed in the production of antimicrobial beta defensins in Figure 3 and bacterial growth in Figure 4B) but may improve the placental barrier defense against *S. agalactiae* infection (Figure 4A).

We then analyzed if exposure to a high glucose concentration in the presence or absence of hypoglycemic treatments modified cytokines secretion in response to *S. agalactiae* infection. As expected, the placental infection challenge significantly induced TNF-α, IL-1β and IL-6 secretion in the explants pretreated with 10 mM glucose (Figure 4C–E). As noted, the infection-dependent inflammatory stimulus was significantly greater than that observed by hyperglycemia alone (50 mM glucose without infection). Unexpectedly for us, the explants pretreated with 50 mM glucose and then infected with *S. agalactiae* exhibited a significantly decreased secretion of TNF-α and IL-6 in comparison with the infected explants incubated with 10 mM glucose (Figure 4C,D). Notably, pretreatment with insulin or metformin did not modify the decreased secretion of TNF-α and IL-6 observed under hyperglycemia. These hypoglycemic treatments results were not significantly different from the those obtained with infected- 50 mM glucose-treated explants. Notably, the infection with *S. agalactiae* significantly stimulated IL-1β secretion, irrespective of the glucose concentration and the presence of insulin or metformin (Figure 4E). Therefore, and contrary to what we expected, a double inflammatory stimulus by severe hyperglycemia and *S. agalactiae* infection resulted in a weaker inflammatory profile, despite each one of these insults alone led to increased TNF-α and IL-6 production in comparison to normoglycemia. 

After the experiments with *S. agalactiae* infection in our model, we tested *E. coli*, another clinically relevant bacteria during pregnancy. Gram-negative bacteria stains as red/pink structures, and we observed that 50 mM glucose induced a broad bacterial invasion of placental villi in comparison with the explants incubated with 10 mM glucose (Figure 5A). In fact, the syncytial layer was an effective barrier against *E. coli* infection in the 10 mM glucose-treated explants, as most of the bacteria were contained at this level. Contrastingly, this barrier was weakened by the presence of 50 mM glucose. As a result, bacteria readily crossed the syncytiotrophoblast layer and penetrated the villous mesenchyme. Interestingly, under this condition, bacteria seemed to also invade fetal capillaries. On the other hand, insulin and metformin acted as good fortifiers of the innate defense against *E. coli*, because they helped to limit bacterial invasiveness into the mesenchyme and capillaries.

Next, we quantified the bacterial growth in the culture media of these *E. coli*-infected explants. As shown in Figure 5B, the culture media of the placental explants pre-incubated with 50 mM glucose had significantly higher *E. coli* counts in comparison with that from the explants grown in 10 mM glucose at 4 h of infection; this difference was lost at 8 h post-infection. Interestingly, insulin and metformin pretreatments in hyperglycemic media significantly diminished *E. coli* growth compared with 50 mM glucose. This behavior agrees with the observed deeper invasiveness of *E. coli* in the presence of 50 mM glucose, as well as the lesser invasion of the villous trees promoted by insulin and metformin (Figure 5A).

Finally, we evaluated the production of pro-inflammatory cytokines in this double challenged placental model. As expected, infection significantly induced IL-1β, TNF-α and IL-6 placental secretion (Figure 5C–E). Again (as observed with *S. agalactiae* experiments), the double hit of hyperglycemia plus infection modified the placental proinflammatory profile. In this case, the explants pretreated with 50 mM glucose and then infected with *E. coli* exhibited a significantly decreased secretion of IL-1β, whereas IL-6 and TNF-α were not modified. Pretreatment with insulin or metformin did not modify the decreased secretion of IL-1β observed under hyperglycemia, and resulted in a significantly lesser secretion in comparison with the infected placental explants exposed to 10 mM glucose. Additionally, these hypoglycemic treatments were not significantly different from the infected explants incubated under 50 mM glucose (Figure 5C). 

All these results point to hyperglycemia as a serious and deleterious scenario which hampers the production of pro-inflammatory cytokines in response to an infection, which may vulnerate maternal host defense. 

## 3. Discussion

In this manuscript, we established experimental in vitro conditions to preserve both the hyperglycemic and the inflammatory states associated with GDM and hyperglycemia in pregnancy. This model, consisting of cotyledon explants exposed to 50 mM glucose, resulted in a significantly induced secretion of TNF-α, IL-1β and IL-6, and more abundant glycogen deposits, without compromising cellular viability. All these changes concur with the well-described morpho-functional alterations of GDM placentae [5,47,52,53]. Multiple models of hyperglycemia have been developed in established cell lines or primary cultures of human placenta. As the control, glucose concentrations fluctuating from 5 mM to 11 mM have been used to recreate a non-diabetic non-inflammatory state in human placental cell lines [54,55], while concentrations ranging from 20 mM to 50 mM have been frequently used to reproduce an environment similar to gestational diabetes mellitus [6,56,57]. In this study, only the treatment with 50 mM glucose effectively induced the secretion of TNF-α and IL-6, while IL-1β was significantly upregulated starting from 35 mM glucose. 

It is worth mentioning that 10 mM glucose is above the cut-off repeated fasting plasma glucose levels for diagnosing GDM (range between 5.1 and 6.9 mM) [58]; however, this concentration is below or in accordance with the standard glucose concentration for the culture of placental explants, usually cultivated in DMEM-HG, DMEM/F12 or RPMI-1640 medium (containing 25 mM and 11.1 mM glucose, respectively) [59]. Notably, even if glucose ~11 mM has been used as an experimental control for hyperglycemic assays in cultured trophoblasts [54,55,60,61,62], herein, we evaluated proinflammatory cytokines secretion in cultured explants under 5 and 10 mM glucose concentrations, and we did not observe a significant difference between both conditions (Appendix A). In addition, the concentrations of insulin and metformin in our study were chosen based on previous experiments performed in tissues of the maternal–fetal interface [6,61,63,64]. Hence, and as a limitation of this study, the glucose, insulin and metformin concentrations tested herein are more in line with experimental conditions than they are with clinical settings. 

Notably, the inflammatory stimulus evoked by hyperglycemia was of a lesser degree than that aroused by LPS, a bacterial endotoxin widely used to promote an acute inflammatory response in both cellular and animal models [65,66,67,68]. Therefore, this experimentally induced inflammation coincides with the low-grade chronic metainflammation observed in diabetic patients, which has been associated with negative short- and long-term adverse effects [69]. Although chronic hyperglycemia in GDM is the main trigger for metainflammation, other factors such as lipotoxicity, endotoxemia or self-nucleic acids may also be involved [70,71], which are probably co-acting in this model. 

On the other hand, several reports have indicated that hyperglycemia weakens innate defense. For instance, we know that hyperglycemia strengthens diverse bacterial and fungal pathogenic mechanisms [72,73], impairs B cells function [74] and dysregulates the synthesis of proteins related to innate immunity defense such as antimicrobial defensins, inflammatory cytokines and chemokines [2,11,75]. Indeed, immune defense against infection involves complex mechanisms and, undoubtedly, a critical one is the synthesis of antimicrobial peptides [76]. Among them, HBDs are the most studied group of microbicidal peptides in the maternal–fetal interface [77,78,79,80]. In vitro and in vivo experiments have demonstrated that a high glucose concentration in sera or culture media diminished the production of HBDs [81,82,83,84]. Accordingly, our results showed that a high glucose environment also diminished the synthesis of placental defensins. In addition to HBDs, visfatin was also downregulated by hyperglycemia. Experimental evidence points out that this adipokine is an enhancer of the synthesis of diverse antimicrobial peptides, including HBDs, cathelicidin and psoriasin [50,51]. Therefore, visfatin reduction in hyperglycemic placenta could be an additional factor explaining the compromised innate response in this model and in GDM pregnancies. Altogether, our results suggest that hyperglycemia impairs the ability of the placenta to respond to an infectious challenge due, at least in part, to a lower synthesis of HBDs and probably other antimicrobial peptides. Further studies are needed for additional mechanistic insights. Is important to note that, in this model, we could not evaluate the role of immune cells and decidua as highly important producers of HBDs and other antimicrobial peptides at the feto-maternal interface [85,86,87], and therefore, these results do not reproduce the entire response occurring at the maternal–placental–fetal interface in vivo. When we analyzed if insulin and metformin modulated the synthesis of inflammatory markers in our placental model, we found that both hypoglycemics helped to reduce the hyperglycemic-induced secretion of TNF-α, IL-1β and IL-6. Accordingly, diminished levels of total NF-κB or its phosphorylation have been described after insulin or metformin treatment in diverse in vitro and in vivo models [88,89,90,91,92]. This action prevents the NF-κB complex from being internalized in the nucleus, which in turn blocks the transcription of genes associated with inflammation. More studies should be carried out in the future to elucidate whether insulin or metformin inhibit the activation of the main promoters of inflammation, especially NF-κB, in the hyperglycemic placenta.

Regarding the regulation of HBDs by insulin and metformin treatment, unexpected results were obtained. We had hypothesized that treating placental explants with both hypoglycemic agents would help strengthen the placental innate defense by increasing the synthesis of HBDs. As previously described, several reports supported that insulin and metformin play an inductive role in the expression of antimicrobial peptides. These studies were undertaken in animal models (diabetic rats and worms) or by in vitro approaches (kidney cell lines, pneumocytes) [38,40,81,93,94]. However, all of them are far from representing the morphological and functional characteristics of the human GDM-placenta. Herein, we demonstrated that neither treatment with insulin nor metformin helped to revert the hyperglycemic-dependent reduction in HBDs production by the human placenta. Interestingly, in a previous diabetic animal model [81], insulin was effective in restoring BD1 expression in the rat kidney; however, the low expression of BD1 due to diabetes could not be restored after insulin treatment in either the lung or brain of the same rats. The latter supports the existence of tissue-specific inductive and repressive mechanisms controlling the beta defensins synthesis in response to insulin.

In accordance with the lower HBDs production, severe hyperglycemia has been pointed out as a strong associative factor related with infections [75,95,96]. During pregnancy, an infectious process into the uterine cavity represents a major challenging condition that endangers the immune privilege of the maternal–fetal unit, increasing the risk of the premature rupture of membranes and preterm birth [69]. In particular, GDM women with urinary or cervicovaginal infections are in an even more vulnerable position because they have to confront two immune insults: one due to hyperglycemia and the second due to infection. Although this combined scenario is very common in the clinical practice [8,10,14], it has been scarcely studied in experimental biomedical approaches. Therefore, we decided to study the response to pathogen infection in hyperglycemic cotyledon explants by evaluating bacterial growth and invasiveness, as well as the inflammatory cytokines synthesis in response to infection. To our knowledge, this is the first experimental approach addressing the combined scenario of hyperglycemia and infections in the human placenta. 

As expected, the hyperglycemic condition significantly decreased the innate defense capacity of the placenta against both *E. coli* and *S. agalactiae* infection. Severe hyperglycemia favored a higher count and deeper invasiveness of both Gram-positive and Gram-negative bacteria. Interestingly, the explants pretreated with glucose 10 mM were barely positive for *S. agalactiae*, even after 8 h post-infection, whereas hyperglycemia led to abundant bacteria contained at the syncytial barrier. In our model, insulin was effective in limiting *S. agalactiae* adhesion to the syncytial layer, but it could not diminish the extracellular bacterial growth. The latter may imply that insulin does not potentiate the synthesis of beta defensins by the human placenta, although the regulation of other antimicrobials is not discarded. Additionally, we believe that other mechanisms of innate defense could be activated by insulin. For instance, insulin treatment helps strengthen the epithelial barrier function by increasing transepithelial electrical resistance [97], while diminishing the permeability of monolayers through the induction of tight junctions [98,99]. Probably, some of these strategies for cellular defense, alone or concomitant, could be occurring at the placenta, which deserves to be further explored. 

In relation to *E. coli* infection, the precondition with glucose 10 mM favored placenta to contain infection mainly at the syncytiotrophoblast barrier, whereas exposure to hyperglycemia allowed bacteria to cross the syncytial barrier and to profoundly invade the mesenchyme. Interestingly, it appears that *E. coli* had a tropism for capillaries. Notably, insulin and metformin readily decreased *E. coli* counts and showed a better fitted defense barrier to a similar extent as explants pre-exposed to glucose 10 mM. These results agree with the experimental evidence indicating that metformin treatment increases the host resistance to *E. coli* infection, inhibits its microbial adherence, diminishes its response to chemoattracts and compromises its flagellar motility [40,100,101]. 

Interesting results were obtained in relation to cytokine secretion in explants incubated with the double inflammatory scenario. First, and as expected, *E. coli* and *S. agalactiae* infection induced the placental secretion of TNF-α, IL-1β and IL-6. However, the challenge of hyperglycemia plus infection diminished the placental secretion of proinflammatory cytokines in comparison with infected but normoglycemic explants. This behavior resembles an LPS tolerance state. This phenomenon was described in animal and culture models after a low dose of LPS exposition, which then alters the subsequent response to LPS or other inflammatory stimulus by inducing an immunosuppressive state to protect the host against a cytokine-induced damage [102,103]. This immunosuppression is primarily linked with chromatin alterations and diverse epigenetic changes which suppress the transcription of NF-κB target genes and end in the transient silencing of pro-inflammatory genes [104,105,106]. In this manuscript, we describe, for the first time, the same tolerant state in a hyperglycemic placental model, which agrees with an initial pro-inflammatory hit due to hyperglycemia followed by a second hit represented by the endotoxemic challenge (LPS in the case of *E. coli* infection or lipoteichoic acid (LTA) for *S. agalactiae* infection), which results in an immunosuppressive response. 

Furthermore, the attenuated inflammatory response in hyperglycemic placenta was pathogen-specific, with the suppression of TNF-α and IL-6 or IL-1β after *S. agalactiae* and *E. coli* infection, respectively. Thus, the pathogen-specific activation of distinct inflammatory pathways is probably involved in the hyperglycemic placenta. The production and release of IL-1β is regulated by a two-signal NLRP3 inflammasome activation; the first signal is through the stimulation of TLRs and NF-κB activation, leading to the up-regulation of IL-1β protein levels. The second signal is through damage-associated molecular patterns (DAMPs) recognition, the formation of the NLRP3 complex, the activation of caspase 1, which is necessary for the cleavage of pro-IL-1β, and the release of the bioactive mature molecule (reviewed by [107]). The inflammasome can also be activated in a caspase-11-dependent noncanonical pathway, leading to IL-1β release. This noncanonical pathway has been described in Gram-negative bacteria such as *E. coli* infection, but not in Gram-positive pathogens, such as *S. agalactiae.* These caspase-1-dependent and -independent pathways induce pyroptosis, a programmed cell death characterized by an increased inflammation occurring after infection but also in chronic diseases such as diabetes [107,108]. Thus, the decreased levels of IL-1β after *E. coli* infection but not as a result of *S. agalactiae*’s presence in the hyperglycemic placenta could be explained by the inhibition of the inflammasome complex formation or caspase 1 activation. A second possibility could involve the different activation of the pyroptosis pathway by both pathogens.

This lower pro-inflammatory cytokine production may impact placental immunity in different ways. On one side, this could help to avoid an exacerbated inflammatory response at the feto-maternal interface, which may avoid fetal neurodevelopmental damage or preterm birth [109,110,111]. However, on the other hand, the inability to produce enough cytokines to fight an infectious challenge increases the host vulnerability to pathogen virulence and invasiveness, which matches with the observed prolonged maternal infection period and the higher frequencies of cervicovaginal, chorioamnionitis or puerperal infections in GDM patients [8,112]. 

Altogether, these results point out that hyperglycemia associated with GDM induced profound immune-metabolic adaptations in the placenta, resulting in a marked inflammatory profile incapable of restraining a bacterial infection, particularly against *E. coli* and *S. agalactiae*, representing the most frequent Gram-positive and Gram-negative etiologic agents of urinary and cervicovaginal infections. Insulin and metformin exert anti-inflammatory activity and placental protection against *E. coli* and *S. agalactiae* infection.

As a final comment, we recognize that this in vitro model has technical limitations because it only reproduces some placental alterations observed in the GDM placentae (i.e., a proinflammatory state and greater glycogen deposition). However, this model cannot reproduce the distinct placental vascular dysfunctions or complex cellular and tissue interactions that occur in GDM patients or in the in vivo models. Nevertheless, this approach may help in understanding several morphophysiological adaptations that take place in the placenta of GDM women, especially those related to the innate defense response against an infectious challenge including parasites and fungi. As strengths, this model reproduces well-known clinical conditions reported in placentas from GDM patients and also allows for the mimicking of placental infectious processes in a clear timeline.

## 4. Materials and Methods

### 4.1. Ethics Statement

This protocol was approved by the Biosafety, Ethical and Research Committees of the Instituto Nacional de Perinatología Isidro Espinosa de los Reyes (INPer, code number 2018-1-152), Hospital Ángeles México, Hospital Ángeles Lomas (HAL 366/2020) and Hospital Gineco Obstetricia No. 4 IMSS (R-2020-785-043), all of them being in Mexico City. All methodological approaches were conducted according to the Belmont Report. Written informed consent, according to the Declaration of Helsinki, was obtained voluntarily from each mother before the caesarean section.

### 4.2. Sample Collection

Complete placentas were collected from normoevolutive, uncomplicated, term (37.2–40 weeks) pregnant women who gave birth to single newborns, and who attended their cesarean section in the hospitals listed above. According to medical records, the clinical indications for cesarean sections were breech presentation, cephalopelvic disproportion, antecedent of uterine surgery, antecedent of myomectomy and personal maternal decision. The exclusion criteria for this study comprised patients with endocrine, metabolic, infectious and/or other systemic diseases such as hypertension, diabetes mellitus and thyroid, liver or chronic renal diseases. Additionally, patients allergic to penicillin or streptomycin, as well as patients who suffered from cervico-vaginal infections during the third trimester of pregnancy, were excluded. 

### 4.3. Reagents

The DMEM culture media and fetal bovine serum (FBS) were from Biowest (Riverside, MO, USA). Monohydrated dextrose was acquired from JT Baker–Fisher Scientific (Mexico City, Mexico) and was diluted in deionized water to a 1 M solution, filtered and preserved at 4 °C. Human insulin was reconstituted in 10 mM HCl to stock at a 1 mM solution. Metformin hydrochloride was reconstituted in phosphate buffer saline (PBS) to a 1 M stock solution. Both hypoglycemics were stored in single-use working aliquots at −20 °C. The LPS from *E. coli* 055:B5 was diluted in PBS to a stock at 100 μg/mL. Insulin (I2643), metformin (PHR1084) and LPS (L4005) were purchased from Sigma-Aldrich (St. Louis, MO, USA).

### 4.4. Cotyledon Explant Culture and Experimental Procedures

Placental cotyledons were exhaustively washed with sterile 0.9% NaCl, and visible blood clots, blood vessels, decidua and chorionic and basal plates were removed. Three small placental explants (around 3 to 5 mm^3^) were placed into 24-well culture dishes and were incubated in DMEM-supplemented culture media (plus 10% FBS + 1% sodium pyruvate + 1% penicillin/streptomycin) in a humidified incubator at 37 °C and 5% CO_2_–95% air. DMEM low glucose (5 mM) was adjusted with a D-glucose solution (dextrose) to generate culture media with glucose 10, 25, 35 or 50 mM. The glucose dose–response curve was chosen according to previous experiments developed in trophoblasts and other cells to mimic normo- and hyperglycemic environments associated with GDM or diabetes [6,43,44,45,46]. After the set of the dose–response experiments, we chose glucose 10 mM as the control glucose non-inflammatory media, whereas glucose 50 mM was chosen as a severe hyperglycemic inflammatory condition. 

The experimental treatment with insulin or metformin was maintained for 48 h, refreshing them at 24 h. We worked with insulin (50, 100 and 500 nM) and metformin (125, 250 and 500 μM) according to previous studies developed in tissues of the materno–placental–fetal interface [6,61,63,64]. After 48 h of incubation, culture media were frozen until cytokine quantification. 

### 4.5. Quantification of Cytokines and Adipokines by ELISA

After thawing the culture media, we employed an R&D System (Minneapolis, MN, USA) ELISA commercial kit to detect the placental secretion of TNF-α (DY210), visfatin (DY4335), chemerin (DY2324), adiponectin (DY1065) and leptin (DY398). IL-1β and IL-6 were quantified using Peprotech (Minneapolis, MN, USA) ELISA commercial kits (900-K95 and 900-K16, respectively). Assays were performed according to the manufacturer’s instructions. Sigma-Fast OPD was used as the colorimetric substrate (P9187, Sigma-Aldrich. St Louis, MO, USA). Absorbances were read with the microplate spectrophotometer xMark (BIO-RAD. Hercules, CA, USA). At the end of each experiment, the placental explant weight was registered, and the concentration of each analyte was normalized by 1 g of wet tissue. 

### 4.6. Tissue Protein Extraction and Human Beta Defensins Quantification by ELISA

At the end of the experiment, the explants were placed into cold protein lysis buffer (Tris-HCl 20 mM, EDTA 1 mM, Nonidet P-40 0.025%, sodium deoxycholate 1%, SDS 0.1%, NaCl 150 mM, Na_3_VO_4_ 2 mM, NaF 50 mM and protease inhibitor cocktail Sigma-Aldrich P8340, 1:1000) (Sigma-Aldrich. St Louis, MO, USA) and stored at −40 °C until the quantification of beta defensins. The explants were mechanically disrupted using a polytron homogenizer (OMNI International. Kennesaw, GA, USA). Then, the sample lysates were centrifuged at 3500 rpm for 10 min at 4 °C, and the supernatant was collected. The protein content of the tissue lysates was quantified by the Bradford method [113]. The samples were adjusted to load 350 μg of the total placental protein per well for the ELISA procedure. The amounts of HBD-1, HBD-2, HBD-3 and HBD-4 in the tissue lysates were quantified using Peprotech (Minneapolis, MN, USA) ELISA commercial kits (900-K202, 900-K172, 900-K210 and 900-K435, respectively), with an 8 pg/mL detection limit for HBD-1, HBD-2 and HBD4 and a 64 pg/mL detection limit for HBD-3. The assays were performed according to the manufacturer’s instructions. Intratissue beta defensins concentrations were normalized per mg of protein.

### 4.7. Placental Infection with Escherichia coli or Streptococcus agalactiae

The *E. coli* strain used herein was isolated from the blood of a neonate diagnosed with early onset sepsis whose mother was diagnosed with premature rupture of membranes and chorioamnionitis at the INPer. This strain was previously utilized in our lab [79], was identified by means of Vitek^®^ and was confirmed by its 16S rRNA sequencing. The *E. coli* genotype was determined by multiplex PCR (Polymerase Chain Reaction). The B2-phylogroup and the detected virulence genes were *PAI*, *papA*, *fimH*, *ibeA*, *fyuA*, *iutA (aerJ)*, *hlyA* and *traT*; it was therefore a pathogenic strain. On the other hand, *Streptococcus agalactiae* Lehmann and Neumann was obtained from the American Type Culture Collection (ATCC 27956. Rockville, MD, USA); as a control, their beta hemolytic activity was corroborated in every experiment. 

Placental explants were infected after 48 h of incubation with the experimental treatments. The night before infection, *E. coli* or *S. agalactiae* were grown in brain heart infusion broth at 37 °C. The bacterial count was calculated by spectrophotometry (600 nm wavelength), and the infection rate (1 × 10^5^ CFU/mL) was corroborated in every experiment by counting the visible *E. coli* colonies in Miller’s LB agar plates or the visible *S. agalactiae* colonies in blood agar plates plus 5% ram’s blood. Both infections were maintained in DMEM-supplemented media without antibiotics (DMEM antibiotic-free media adjusted to glucose 10 mM + 0.2% lactalbumin hydrolysate + 1 mM sodium pyruvate) for 4 and 8 h. 

A bacteria count of 1 × 10^5^ CFU/ mL in a urinalysis test represents a pathognomonic sign of urinary tract infection [114,115]; therefore, it was chosen for the infection challenge experiments. An aliquot of culture media was used for the CFU count. Briefly, serial 10-fold dilutions were seeded in triplicate into Miller’s LB or blood agar plates. After overnight incubation at 37 °C, the plates with visible and countable CFU were chosen, and the corresponding dilution was registered. The final number of colonies in each treatment was estimated by the following Equation (1).
(1)$$\mathrm{CFU}/\mathrm{mL}=\frac{\left(\mathrm{No}.\;\mathrm{of}\;\mathrm{colonies}\right)\left(\mathrm{Dilution}\;\mathrm{factor}\right)}{\mathrm{Volume}\;\mathrm{of}\;\mathrm{culture}\;\mathrm{well}\;\left(1000\;\mu L\right)}$$

Non-infected explants were also maintained with DMEM-supplemented media without antibiotics as controls. 

### 4.8. Staining Techniques

At the end of the experimental procedure, placental explants were fixed in 10% formalin for at least 24 h. The tissues were embedded in paraffin blocks and cut into 10 µm slices. The slides were deparaffinized and cleared in Histo-Clear (National Diagnostic, Atlanta, GA, USA) and rehydrated through graded concentrations of ethanol in water for staining. The sections were stained for modified Gram (Remel, Lenexa, KS, USA) or for Periodic acid–Schiff stain (PAS). 

Gram dye allowed for the visualization of Gram-negative bacteria in the tissues as pink- to red-colored structures, whereas Gram-positive bacteria are shown as purple- to blue-colored bodies; the tissue was counterstained with yellow picric acid. 

PAS staining is frequently used to detect different sugars in tissues (such as glycogen, glycoproteins, glycolipids or mucins), and it appears as magenta regions; the nuclei were positive for hematoxylin.

### 4.9. XTT Cell Viability Assay

Cell viability was quantified using the Cell Proliferation Kit II XTT (Roche Diagnostics, Basel, Switzerland), according to the manufacturer’s instructions. Briefly, placental explants were incubated for 96 h with culture media adjusted with glucose 10 or 50 mM. On the day of the experiment, the XTT Reagent/Electron Coupling Reagent mixture was added, and the explants were incubated for 2 h at 37 °C in a humidified incubator 5% CO_2_–95% air. The absorbance was read at 450 nm using the microplate spectrophotometer xMark (BIO-RAD. Hercules, CA, USA). Absorbance was corrected to wells with XTT substrate and no tissue.

### 4.10. Statistical Analysis

Depending on the normality data distribution (Shapiro–Wilk test), statistical comparisons were made by parametric One-Way ANOVA followed by Tukey’s post hoc test for multiple comparisons or by a non-parametric Kruskal–Wallis test followed by Dunn’s post hoc test for multiple comparisons. Data are presented as the mean ± standard deviation for normal data or as the median and interquartile range for non-normal data, as indicated in the figure legends. Statistical analyses were performed with GraphPad Prism software version 9.5.0 (San Diego, CA, USA). *p* < 0.05 denotes statistically significant differences.

## 5. Conclusions

Our results demonstrated a useful model for experimentally emulating a hyperglycemic milieu in the term human placenta. In this model, we showed that severe hyperglycemia promotes the placental synthesis of inflammatory cytokines, while it reduces the synthesis of beta defensins. Importantly, hyperglycemia compromised the placental innate defense against *E. coli* and *S. agalactiae* infection, denoted by higher CFU counts and broader bacterial invasiveness. Insulin and metformin treatments were effective anti-inflammatory treatments in hyperglycemic infectious and non-infectious scenarios. In relation to their properties against bacterial infections, insulin limited the invasiveness of both *E. coli* and *S. agalactiae* into placental villous trees and diminished the extracellular *E. coli* growth. On the other hand, the biguanide metformin improved the placental innate defense by reducing *E. coli* counts and their bacterial invasiveness into villous trees. Interestingly, we observed that cotyledons pre-exposed to hyperglycemic media responded with a significantly decreased inflammatory response against a bacterial insult, which may correlate with a cytokine tolerization process and may help to explain the increased maternal vulnerability to bacterial pathogens during hyperglycemic pregnancies. 

## Figures and Tables

**Figure 1 ijms-24-05243-f001:**
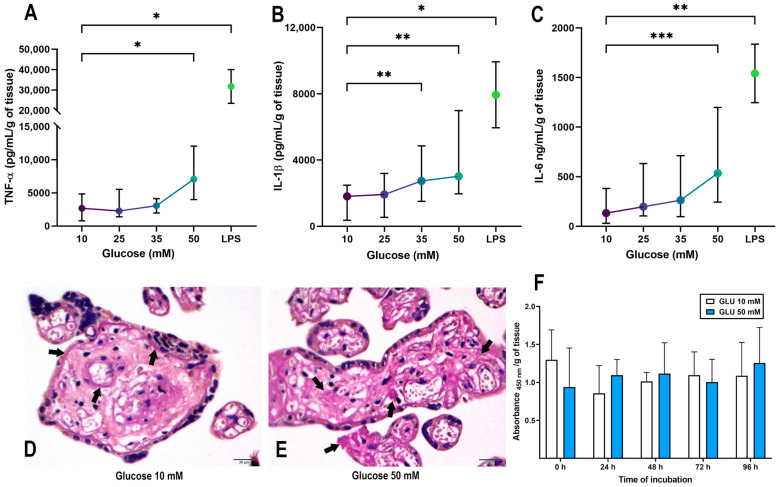
Validation of a hyperglycemic model in human term placenta. The glucose curve (10–50 mM) was probed to evaluate the placental secretion of (**A**) TNF-α, (**B**) IL-1β and (**C**) IL-6. Placental explants were exposed to glucose media for 48 h, and culture media were refreshed at 24 h. Treatment with LPS was applied once at 500 ng/mL. *n* = Eight independent experiments in triplicate. Kruskal–Wallis test followed by Dunn’s multiple comparisons post hoc test. Data are presented as the median (dot) and interquartile range (whiskers). All comparisons were made vs. Glucose 10 mM: *, *p* < 0.05. **, *p* < 0.01. ***, *p* < 0.001. (**D**,**E**) Histological examination of glycogen deposition by PAS staining in placental explants exposed to glucose 10 mM or 50 mM for 48 h. Arrows indicate sites of glycogen deposits. Representative microphotographs from four independent experiments. Scale indicates 20 μm at the bottom right. (**F**) XTT cell viability assay of placental explants incubated for 96 h, with culture media adjusted with glucose 10 mM (white bar) or 50 mM (blue bar). *n* = Three independent experiments in triplicate. Two-way ANOVA (mixed effects analysis) followed by Tukey´s and Sídák’s multiple comparisons test. No significant differences were detected. Data are presented as the mean and standard deviation.

**Figure 2 ijms-24-05243-f002:**
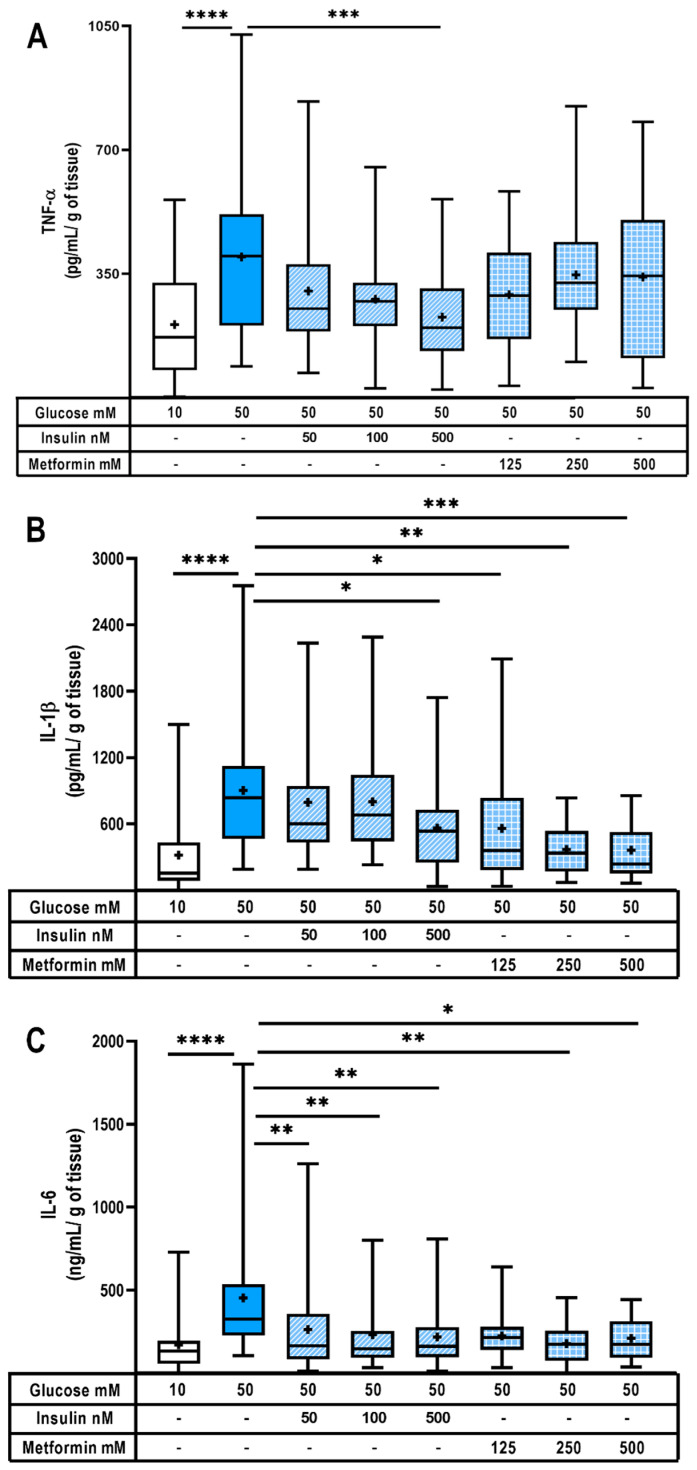
Insulin and metformin diminish pro-inflammatory cytokines in hyperglycemic placenta. Placental explants secretion of (**A**) TNF-α, (**B**) IL-1β and (**C**) IL-6. Explants were exposed to glucose (10 or 50 mM) and coincubated with insulin (50, 100, 500 nM) or metformin (125, 250, 500 μM) for 48 h. Treatments and culture media were refreshed at 24 h. *n* = 10 independent experiments in triplicate. Data are presented as boxes (percentiles 25, 50 and 75) and whiskers (min-max). Symbol + marks the mean. Kruskal–Wallis test followed by Dunn´s multiple comparisons post hoc test. Comparisons were made versus glucose 50 mM: *, *p* < 0.05; **, *p* < 0.01; ***, *p* < 0.001; ****, *p* < 0.0001.

**Figure 3 ijms-24-05243-f003:**
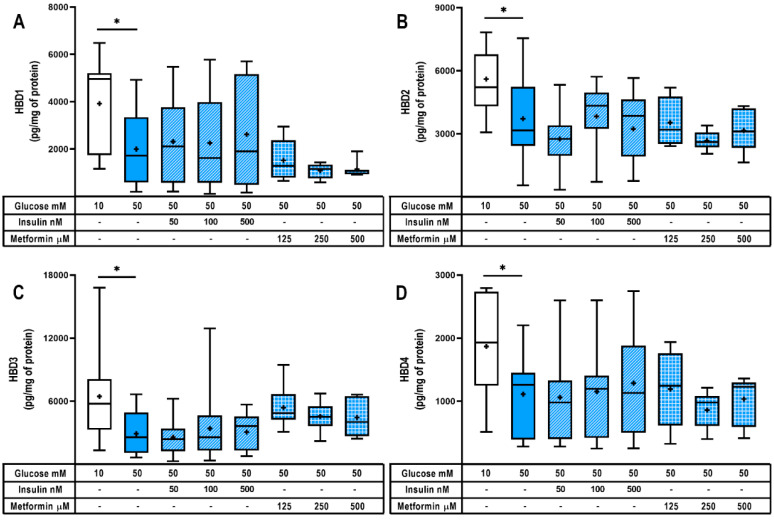
Hyperglycemia diminishes HBDs synthesis in human placenta. Intracellular content of (**A**) HBD1, (**B**) HBD2, (**C**) HBD3 and (**D**) HBD4. Placental explants were exposed to glucose media (10 or 50 mM) and coincubated with insulin (50, 100, 500 nM) or metformin (125, 250, 500 μM) for 48 h. Treatments and culture media were refreshed at 24 h. *n* = Three to five independent experiments in triplicate. Data are presented as boxes (percentiles 25, 50 and 75) and whiskers (min-max). Symbol + marks the mean. Kruskal–Wallis test followed by Dunn´s multiple comparisons post hoc test. Comparisons were made versus glucose 50 mM: *, *p* < 0.05.

**Figure 4 ijms-24-05243-f004:**
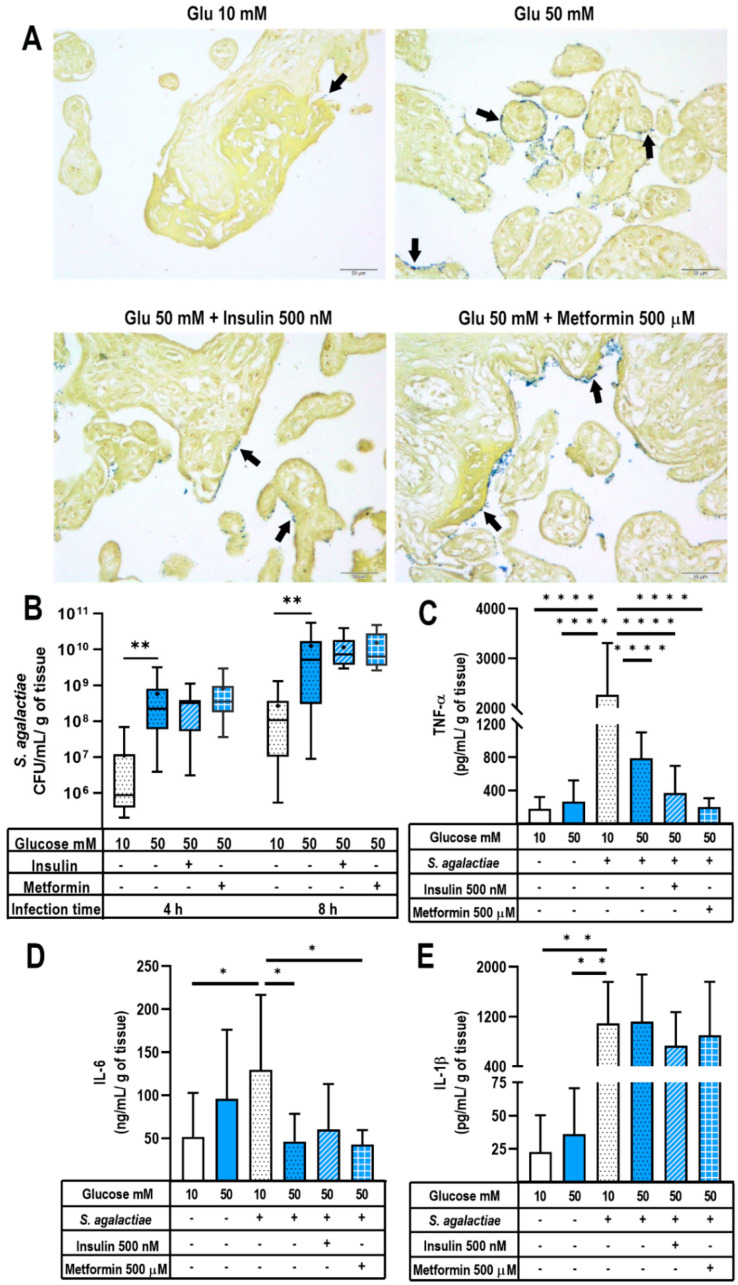
Innate defense against *Streptococcus agalactiae* infection in a hyperglycemic placental model. Placental explants were exposed to glucose media (10 or 50 mM) for 48 h and cocultivated with insulin 500 nM or metformin 500 μM. Then, a constant load of *S. agalactiae* (1 × 10^5^ CFU/mL) was inoculated in antibiotic-free medium. Incubations proceeded at 37 °C for 4 and 8 h. (**A**) Histological examination of *S. agalactiae*-infected placental explants was performed 8 h after tissue inoculation to evaluate the bacterial invasiveness of the villous trees. Arrows point to Gram-positive bacteria contained at the syncytial barrier. Representative microphotographs from three independent Gram-stained experiments are shown. Scale at 50 μm is shown at the bottom right. (**B**) Bacterial growth in the culture media of *S. agalactiae*-infected explants. Explants were pretreated with glucose 10 or 50 mM, insulin 500 nM or metformin 500 μM. An aliquot of the media was used to assess CFU/mL at each follow-up time point. Data are presented as boxes (percentiles 25, 50 and 75) and whiskers (min-max). Symbol + marks the mean. *n* = Five independent experiments in triplicate. Kruskal–Wallis test followed by Dunn´s multiple comparisons post hoc test. **, *p* < 0.01 vs. glucose 50 mM at each time point. (**C**) TNF-α secretion, (**D**) IL-6 secretion and (**E**) IL-1β secretion in *S. agalactiae*-infected or non-infected explants for 8 h. Explants were pretreated with glucose 10 or 50 mM, insulin 500 nM or metformin 500 μM. *n* = Five independent experiments in triplicate. Data are presented as the mean and standard deviation. TNF-α secretion was statistically compared by a Kruskal–Wallis test followed by Dunn’s multiple comparisons post hoc test due to no normal distribution. IL-1β and IL-6 were statistically compared by One-way ANOVA followed by Tukey’s multiple comparisons post hoc test due to normal distribution. All comparisons were made vs. infected Glucose 10 mM: *, *p* < 0.05. **, *p* < 0.01. ****, *p* < 0.0001.

**Figure 5 ijms-24-05243-f005:**
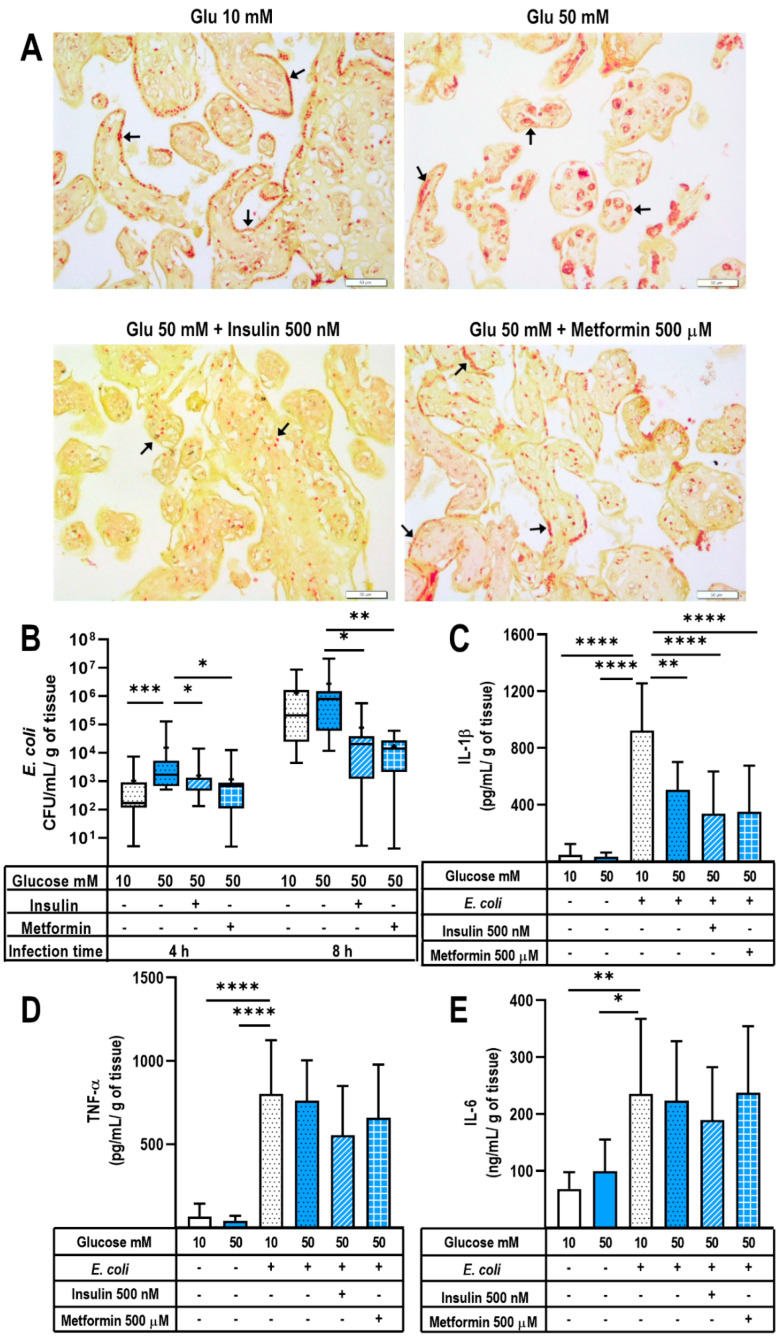
Innate defense against *Escherichia coli* infection in a hyperglycemic placental model. Placental explants were exposed to glucose media (10 or 50 mM) for 48 h and cocultivated with insulin 500 nM or metformin 500 μM. Then, a constant load of *E. coli* (1 × 10^5^ CFU/mL) was inoculated in antibiotic-free medium. Incubations proceeded at 37 °C for 4 and 8 h. (**A**) Histological examination of *E. coli*-infected placental explants was performed 4 h after tissue inoculation to evaluate the bacterial invasion of the villous trees. Arrows point to Gram-negative bacteria. Representative microphotographs from three independent Gram-stained experiments are shown. Scale at 50 μm is shown at the bottom right. (**B**) Bacterial growth in the culture media of *E. coli*-infected explants. Explants were pretreated with glucose 10 or 50 mM, insulin 500 nM or metformin 500 μM. An aliquot of the media was used to assess CFU/mL at each follow-up time point. Data are presented as boxes (percentiles 25, 50 and 75) and whiskers (min-max). Symbol + marks the mean. *n* = Five independent experiments in triplicate. Kruskal–Wallis test followed by Dunn´s multiple comparisons post hoc test. *, *p* < 0.05. ****, *p* < 0.0001 vs. glucose 50 mM. (**C**) IL-1β secretion, (**D**) TNF-α secretion and (**E**) IL-6 secretion in *E. coli*-infected or non-infected explants for 8 h. Explants were pretreated with glucose 10 or 50 mM, insulin 500 nM or metformin 500 μM. *n* = Five independent experiments in triplicate. Data are presented as the mean and standard deviation. One-way ANOVA followed by Tukey’s multiple comparisons post hoc test. All comparisons were made vs. infected glucose 10 mM: *, *p* ≤ 0.05. **, *p* < 0.01. ***, *p* < 0.001. ****, *p* < 0.0001.

**Table 1 ijms-24-05243-t001:** Clinical data of mothers and newborns.

Clinical Parameter	Mean ± SD (*n* = 35)	Range (Min–Max)
Maternal age (years)	30.6 ± 5.1	19–40
Pre-gestational BMI (kg/m^2^)	25.4 ± 3.3	18.6–30.5
Number of pregnancies	2 ± 1	1–5
Parity (Primiparous/Multiparous) (%/%)	9/26	25.7%/74.3%
Gestational age (weeks)	38.4 ± 0.6	37.2–40
SBP (mm Hg)	112 ± 10	99–140
DBP (mm Hg)	75 ± 9	60–107
Maternal weight gain (kg)	11.1 ± 4.1	3.5–20
NB weight (kg)	3.1 ± 0.3	2.5–3.9
NB weight centiles	49 ± 25	15–97
NB length (cm)	49 ± 1	47.5–53
NB length centiles	61 ± 25	15–98
NB head circumference (cm)	35 ± 1	32.5–37.5
NB head circumference centiles	77 ± 24	17–99
APGAR 1 min	8.4 ± 0.5	7–9
APGAR 5 min	9.2 ± 0.4	9–10
Placenta weight (g)	536 ± 122	312.4–728.8
NB sex (male/female) (%/%)	17/18	48.5%/51.5%

APGAR: acronymus for the score system of the physical signs of the newborn (Activity, Pulse, Grimace, Appearance and Respiration). BMI: body mass index. DBP: diastolic blood pressure. SBP: systolic blood pressure. NB: newborn. Placenta weight was registered without fetal membranes and the umbilical cord. Sex- and gestational age- specific centiles for weight, length and head circumference were calculated with the newborn size calculator of the INTER GROWTH-21st Project, available online [41,42].

## Data Availability

The data used to support the findings of this study are available from the corresponding author upon request.

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
