# Peer review of "High Glucose Promotes Inflammation and Weakens Placental Defenses against E. coli and S. agalactiae Infection: Protective Role of Insulin and Metformin"

_ijms, 2023, doi:10.3390/ijms24065243_

Round 1
Reviewer 1 Report
This is an interesting article; however, there are details that should be reviewed.
Lines 50-53: Reference 1 does not support the text of these lines as it is a review of DM2 and obesity outside pregnancy.
Lines 55-56: References 3-4 do not mention an altered innate immune cell infiltration in decidua of women with GDM.
Lines 58-59: References 6-7 do not support the observation that hyperglycemia induces TNF-α secretion in the aforementioned cell line.
Lines 59-62: Referring to the secretion of pro-inflammatory cytokines caused by hyperglycemia in trophoblast cell line, the authors mention that “However, this sustained inflammatory status does not result in an effective host innate defense against infections…” (sic); but do not provide a reference to support that statement, instead they seem to extrapolate what was observed in the studies of references 8-10, which by the way none of them were performed in trophoblast.
Lines 69-60: Misspelled "counseling".
Lines 77-79: It seems to me that the insulin receptor was not blocked but deleted in intercalated cells-specific IR knockout mice.
Lines 79-80: Reference 40 does not mention that insulin treatment promotes the synthesis of hBD-2 and cathelicidin.
Table 1: Express the number of pregnancies as percentages of parity (primiparous and multiparous). The meaning of BMI and APGAR is missing at the bottom of the table.
Lines 117-119: The authors mention that “Notably, the achieved glucose-dependent increase in inflammatory cytokines was not as high as that observed with lipopolysaccharide (LPS), which represents in this model an acute stimulus for inflammation…” (sic); but, they do not provide data to support this statement. In the event that the authors are simply describing the findings of reference 5, it is better to point this out in the discussion section or be clearer because it would appear that they are comparing their proinflammatory cytokine results with those of reference 5.
Supplementary table 1: A comma is missing in the second row of visfatin concentrations.
Line 318: In this section it is necessary 1) to discuss how close to reality the concentrations of glucose, insulin, and metformin used in the experiments are to reality; 2) to avoid extrapolating the findings exclusively to GDM, since in reality the model used is of hyperglycemia in pregnancy, and 3) to make it very clear that the results of the study only point to the placental component of the response against infectious agents and not to the entire response that occurs at the maternal-placental interface, especially when talking about the levels of HBDs, since innate immune cells are important producers of these molecules.
Lines 340-358: The authors should note that their findings related to innate defense (HBD production and bacterial growth) at the maternal-placental interface should be taken with caution because they did not consider the role of decidual immune cells; in fact, they probably eliminated most of them when preparing the explants.
Line 473: What were the indications for cesarean section?
Lines 500-502: Does a 50 mM concentration of glucose really represent a DMG model? In that sense, the authors should interpret and discuss their findings carefully. It is not enough just to mention that this concentration has already been used in other studies.
Why does using these glucose concentrations represent a model of GDM? Could it not also be a model of DM2 in pregnancy? The article does not provide a rationale for considering the model as exclusive for GDM and, therefore, the title exceeds the true scope of their findings.
Line 585: I suggest calculating the effect size (with confidence intervals calculated with bootstrap resampling) to determine how relevant your findings are. On the other hand, was there no statistical control for confounding variables such as maternal age, weight gain, BMI, gestational weight gain, and NB weight? It will be necessary to explain the regression models to be used.
Author Response
Please see the attachment pdf file titled "Response letter to reviewer1. R1"

Reviewer 2 Report
The manuscript entitled “High glucose weakens placental defenses against E. coli and S. agalactiae infection: role of insulin and metformin treatment in gestational diabetes mellitus” by Jiménez-Escutia et al. provides novel findings of inflammatory response in a GDM-like model in placental explants. Moreover, the authors evaluated the anti-inflammatory potential of insulin and metformin under hyperglycemic infectious and non-infectious scenarios. Interestingly, they report that the double challenge of high glucose and infection provokes a pathogen-specific attenuated placental inflammatory response in the hyperglycemic condition. These findings are highly relevant for several aspects of placental pathological changes in pregnancies complicated by GDM. The paper is well written, the experimental approaches are appropriate, and the data representation is adequate. Nonetheless, I have several concerns, particularly with the design of ex vivo experiments and the discussion of the “double inflammatory scenario”. Should the authors address these remarks, I consider that the manuscript would be suitable for publication in IJMS.
Major comments:
- Explant cultures were maintained in normal oxygen conditions, which represents hyperoxia for placental tissue. One previous report in human first-trimester trophoblast cell line Sw.71 (DOI: 10.1111/aji.13044) has shown that oxygen level also affects the way the cells respond to hyperglycemic conditions. The authors should comment on the choice to do experiments at 21% O2 rather than at 8% O2 (normoxia for the placenta).
- Have the authors considered the effect of osmolar pressure when working with such a high concentration of glucose? Were experiments with osmolar controls performed?
- According to the latest recommendation from the World Health Organization (WHO), the cut-off repeated fasting plasma glucose levels for diagnosing GDM range between 5.1 - 6.9 mM. Hence, I have concerns about using the 10 mM glucose as “control” in this study – this is already a hyperglycemic state. Moreover, while the results presented here with 50 mM glucose are interesting, these levels of glucose are highly unlikely in clinical settings. The authors should discuss this as a limitation of the study.
- Similar comment is also in the case of metformin, where the lowest concentration used is 125 µM, and most subsequent experiments are conducted with 500 µM. These levels are far from the achievable plasma concentrations for metformin (which range around 10 µM in pregnant women - DOI: 10.1124/dmd.109.031245).
- Why was the 48-hour incubation period chosen? Did the authors perform a time-dependency curve?
- Viability assays were performed only for glucose - Do the authors have data on explant viability in presence of insulin and metformin for 48 hours?
- Considering that a high n number was used in most experiments – did the authors try to evaluate potential sex-dependent effects? This could maybe help explain the high variability between biological replicates.
- Lines 281 – 283: Do the authors have an explanation that the difference between 10 and 50 mM on E coli growth is lost at 8 hours? And what was the rationale for choosing the 8-hour exposure to evaluate cytokine release in subsequent experiments (Figure 5C-E)?
- Lines 445 – 446: It is interesting that there is a pathogen-specific attenuated inflammatory response in the hyperglycemic placenta, denoted by different cytokines being affected. This could be related to the pathogen-specific activation of distinct inflammatory pathways in the placenta. The authors should discuss this.
Author Response
Please see the attachment pdf file titled "Response letter to reviewer2. R1"

Round 2
Reviewer 2 Report
The authors have addressed all the issues raised during the first round of reviews. Only minor remark - in Figure 1 (A, B, C) the units in the Y axis are pg/g tissue, whereas, in the rest of the experiments with cytokine release in the supernatant, they are pg/ml/g tissue. The authors should unify the units.